# Three-Dimensional Co-Culture System of Human Osteoblasts and Osteoclast Precursors from Osteoporotic Patients as an Innovative Model to Study the Role of Nutrients: Focus on Vitamin K2

**DOI:** 10.3390/nu13082823

**Published:** 2021-08-17

**Authors:** Domitilla Mandatori, Letizia Penolazzi, Letizia Pelusi, Elisabetta Lambertini, Francesca Michelucci, Annamaria Porreca, Pietro Cerritelli, Caterina Pipino, Angelo Di Iorio, Danilo Bruni, Marta Di Nicola, Roberto Buda, Roberta Piva, Assunta Pandolfi

**Affiliations:** 1Department of Medical, Oral and Biotechnological Sciences, University “G. d’Annunzio” Chieti-Pescara, 66100 Chieti, Italy; domitilla.mandatori@unich.it (D.M.); letizia.pelusi@unich.it (L.P.); annamaria.porreca@unich.it (A.P.); caterina.pipino@unich.it (C.P.); marta.dinicola@unich.it (M.D.N.); 2Center for Advanced Studies and Technology (C.A.S.T.), University “G. d’Annunzio” Chieti-Pescara, 66100 Chieti, Italy; 3Department of Neuroscience and Rehabilitation, University of Ferrara, 44121 Ferrara, Italy; maria.letizia.penolazzi@unife.it (L.P.); elisabetta.lambertini@unife.it (E.L.); 4Department of Medicine and Aging Sciences, University “G. d’Annunzio” Chieti-Pescara, 66100 Chieti, Italy; francesca.michelucci1@gmail.com (F.M.); al1993@hotmail.it (P.C.); angelo.diiorio@unich.it (A.D.I.); danilo.bruni@unich.it (D.B.); roberto.buda@unich.it (R.B.)

**Keywords:** osteoporosis, bone health, osteoblasts, osteoclasts, 3-D culture, vitamin K2, personalized medicine

## Abstract

Several natural compounds, such as vitamin K2, have been highlighted for their positive effects on bone metabolism. It has been proposed that skeletal disorders, such as osteoporosis, may benefit from vitamin K2-based therapies or its regular intake. However, further studies are needed to better clarify the effects of vitamin K2 in bone disorders. To this aim, we developed in vitro a three-dimensional (3D) cell culture system one step closer to the bone microenvironment based on co-culturing osteoblasts and osteoclasts precursors obtained from bone specimens and peripheral blood of the same osteoporotic patient, respectively. Such a 3-D co-culture system was more informative than the traditional 2-D cell cultures when responsiveness to vitamin K2 was analyzed, paving the way for data interpretation on single patients. Following this approach, the anabolic effects of vitamin K2 on the osteoblast counterpart were found to be correlated with bone turnover markers measured in osteoporotic patients’ sera. Overall, our data suggest that co-cultured osteoblasts and osteoclast precursors from the same osteoporotic patient may be suitable to generate an in vitro 3-D experimental model that potentially reflects the individual’s bone metabolism and may be useful to predict personal responsiveness to nutraceutical or drug molecules designed to positively affect bone health.

## 1. Introduction

Osteoporosis is one of the major skeletal disease characterized by loss of bone mass, micro-architectural deterioration of bone tissue, and its consequent fragility [1]. This causes an increase of osteoporotic fractures, which have a negative impact on quality of life in high-risk populations [2,3]. Recently, the attention given to lifestyle, a healthy diet, and bone health has grown and several studies have highlighted a strict association between a correct nutritional status and bone outcomes [3,4,5].

Among all possible osteo-protective compounds, vitamin K2 is noteworthy [6], a family of different isoforms, known as menaquinones (MKs), which differ from each other by the number of isoprenoid units in the side chain (*n* = 1–14), and in the last few years, they have received attention for their positive effects in bone metabolism [7,8,9,10,11]. Although its beneficial role in decreasing bone loss and risk of osteoporotic fractures has been established [12,13,14,15], additional studies are needed to deeply understand if vitamin K2 intake would provide a valid support to prevent and treat osteoporosis or if some people may be refractory to its effects.

In this regard, in the last years, researchers have focused their studies on the concept of specific patient-response medicine. This aims not to predict what percentage of patients will respond to a particular pharmacological or natural intervention/treatment, but rather to identify which subjects are likely to respond or not [16,17]. This approach, besides the importance of the conventional clinical studies, relies on the implementation of in vitro approaches. Among these, the use of 3-D culture systems composed of autologous cells is emerging. Indeed, mimicking the in vivo microenvironment, 3-D culture systems could overcome the limits of conventional but not very informative 2-D monolayer cultures and the long process of pre-clinical animal studies [4,18,19]. Regarding bone tissue, the employment of 3-D cell-based models might provide significant support in the study of bone remodeling molecular mechanisms, allowing investigation of the pathophysiology of bone diseases and healing.

In this regard, we previously demonstrated the feasibility of obtaining a 3-D bone construct (3D-DyC) by using a direct cellular interaction between osteoblasts (OBs) and osteoclasts (OCs) co-cultured in a dynamic flow bioreactor [20]. This 3-D scaffold-free approach was further employed to assess the ability of vitamin K2 in inducing human mesenchymal stem cells (MSCs) to perform osteogenic differentiation. Of note, we found that it promoted bone aggregates’ formation from osteo-committed MSCs, emphasizing how this approach could be used to test the efficacy of candidate compounds for the prevention and treatment of bone disease, such as osteoporosis [21]. However, currently, there are no co-culture systems based on osteoblasts and osteoclast precursors obtained from the same osteoporotic patient.

Thus, our aim was to generate an in vitro autologous 3-D bone constructs reproducing the bone microenvironment to test the efficacy of vitamin K2 (Menaquinone-4, MK-4) in improving the functions of osteoblasts isolated from osteoporotic patients.

Additionally, growing attention has turned to the role of bone turnover markers (BTMs), which can be classified into two large groups: bone formation (bone alkaline phosphatase, procollagen type I N-terminal propeptide, carboxy-terminal propeptide of type 1 procollagen, and undercaboxylated osteocalcin) and bone resorption markers (amino- and carboxy terminal cross-linked telopeptide of type I collagen) [22,23,24]. BTMs can be used as potential predictors of bone disease [25] and a possible correlation with data in vitro from the experimental model that we developed could help to investigate whether some subjects may or may not be responsive to vitamin K2. Therefore, in order to carry out a preliminary approach versus the concept of patient-oriented medicine, our further challenge was to analyze the vitamin K2 responsiveness of each patient-derived 3-D bone cellular aggregates and then, to correlate the 3-D in vitro data with the BTM serum levels measured in the osteoporotic patients.

## 2. Materials and Methods

### 2.1. Chemicals

Vitamin K2 powder as Menaquinone-4 isoform (MK-4; Ibersan-Santiveri s.r.l., Forlì, Italy) was re-suspended in dimethyl sulphoxide (DMSO) at a 10 mM concentration and stored at −20 °C. DMSO, ascorbic acid-2-phosphate, β-glycerophosphate, dexamethasone, Alizarin Red S (ARS), paraformaldehyde, Triton X-100, pronase, 4′,6-diamidino-2-phenylindole (DAPI), and tartrate-resistant acid phosphatase kit (TRAP, no. 386) were purchased from Sigma-Aldrich (Saint Louis, MO, USA). Trizol^®^ reagent, High Capacity cDNA Reverse Transcription Kit, TaqMan Gene Expression assays, and Universal Master mix were purchased from ThermoFisher Scientific (Waltham, MA, USA). Vectastain ABC and the DAB (3,3′-Diaminobenzidine) solution were obtained from Vectorlabs (Burlingame, CA, USA).

### 2.2. Patients and Clinical Parameters

Ten consecutive patients (8 female and 2 male) with an osteoporosis-related femoral neck fracture, showing a T-score value less than −2.5 on previously performed bone mineral densitometry, were randomly enrolled in the Orthopaedics and Traumatology Clinic of “Santissima Ss. Annunziata” Hospital (Chieti, Italy) and were included in the present study. All patients sustained a fall from a walking-height distance and the fracture was produced by a low-energy traumatic mechanism. Demographics and clinical parameters are shown in Table 1.

Patient exclusion criteria were: (1) chronic kidney disease (end-stage); (2) progressing neoplastic diseases; (3) chronic autoimmune diseases, with the exception of Hashimoto’s thyroiditis and Graves’ disease in remission; (4) inflammatory diseases (rheumatoid arthritis, inflammatory bowel disease); (5) chronic liver disease; (6) chronic therapy (>3 months) with oral and parenteral intake of glucocorticoids and thiazolinediones; and (7) therapy with oral anticoagulant drugs, such as warfarin. All procedures were in agreement with the Declaration of Helsinki principles and with the ethical standards of the Institutional Committee on Human Experimentation (Reference Number: N°24_22.10.2020). Following the protocol approval by the Institutional Review Board, signed informed consent was obtained from each participating subject to collect human biological samples.

On the first day of hospitalization, peripheral blood samples (5 mL) were collected to obtain serum (centrifugation for 10 min at 3000 rpm). In addition to these, 20 mL of citrated blood samples were used to isolate peripheral blood mononuclear cells (PBMCs).

Bone fragments were collected during surgery from the fractured head-to-neck junction after removal of the femoral head. All samples were immediately sent to the laboratory for prompt harvesting.

### 2.3. BTMs Serum Markers

ELISA kits were used to measure the serum levels of human bone Alkaline Phosphatase (bALP; Cusabio, CSB-E09033h, Houston, TX, USA), procollagen I N-terminal Peptide (PINP; Cusabio, CSB-E11226h), carboxyterminal propeptide of type I procollagen (PICP; Cusabio, CSB-E08079h), cross linked N-telopeptide of type I collagen (NTX; Cusabio, CSB-E09233h), and cross linked C-telopeptide of type I collagen (CTX; Novusbio, NBP2-69073, Centennial, CO, USA).

EIA kits were used to measure serum levels of human osteocalcin in the undercaboxylated (unOC) and carboxylated (cOC) form (Takara, Shiga, Japan, MK-118 and MK-111, respectively).

### 2.4. Cell Cultures

After phosphate-buffered solution (PBS) infusion to remove bone marrow, bone fragments were placed in Petri dishes with control medium (CTRL) composed of Dulbecco’s modified Eagle Medium low glucose (DMEM L-GLU, Gibco-Life Technologies, Waltham, MA, USA), 10% fetal bovine serum (FBS, Gibco-Life Technologies), 1% L-glutamine (L-GLU), and 1% of penicillin/streptomycin (P/S) (Sigma-Aldrich). Following 10 days of culture (5% CO_2_ and 37 °C), bone fragments were removed. Isolated cells through spontaneous migration were characterized for their morphology (phase contrast microscopy and Phalloidin staining), ability to deposit mineral matrix (Alizarin red staining), and expression of the following specific bone markers: alkaline phosphatase (ALP), Runt-related transcription factor 2 (Runx2), osteocalcin (OC), osteopontin (OPN), and collagen type 1 alpha (COL1a1). Furthermore, for each patient, 2 × 10^6^ bone cells were cryopreserved in fetal bovine serum (FBS, 90%) and DMSO (10%). For 2-D experiments, hOBs were seeded in six-well plates (50,000 cells/well) and grown in osteogenic medium (OM) composed of CTRL medium supplemented with 0.05 mM of ascorbic acid 2-phosphate, 10 of mM of β-glycerophosphate, and dexamethasone 100 nM in the presence or absence of vitamin K2 (menaquinone-4 isoform, MK-4, 10 μM). MK-4 treatment was repeated every 24 h for 14 days.

For each patient, fresh mononuclear cells (PBMCs) were isolated from 20 mL of peripheral blood, which were separated by Histopaque^®^-1077 (Sigma-Aldrich). PBMCs were resuspended in 800 μL of a mixture of 90% fetal calf serum (FCS)/10% DMSO (Sigma-Aldrich) for cryopreservation. For recovery of frozen cells, the cryovials were rapidly thawed in a 37 °C water bath, and the cell suspension was gradually diluted to 1:10 in Roswell Park Memorial Institute (RPMI) 1640 plus 1% FCS. hMCs were purified from PBMCs by adhesion selection on polystyrene plates: 1 × 10^6^ PBMCs/cm^2^ were plated in T-25 culture flasks, allowed to settle for 4 h at 37 °C, and then rinsed to remove non-adherent cells (lymphocytes, platelets, red blood cells, polymorphonuclear cells). The purity of the hMCs population was verified by cytometric analysis, by using Fluorescein isothiocyanate (FITC) conjugated anti-human CD14 antibody (ImmunoTools GmbH, Friesoythe, Germany). The analysis was performed using FACSVerse (BD Bioscences, San Jose, CA, USA) and the FACSSuite software (BD Bioscences, San Jose, CA, USA). Only samples CD14 positive ≥ 95% were used for the experiments. In order to confirm the ability of isolated hMCs to differentiate into mature osteoclasts (hOCs), M-CSF (25 ng/mL) and RANKL (30 ng/mL) (PeproTech EC Ltd., London, UK) were added to the culture medium. After 10 days, TRAP staining was carried out with the Acid Phosphatase Leukocyte (TRAP) Kit no. 386 (Sigma-Aldrich) according to the manufacturer’s protocol. The presence of Nuclear factor of activated T-cells (NFAT; #sc-13033, Santa Cruz Biotechnology—Dallas, TX, USA) and Cathepsin K (#sc-48353, Santa Cruz Biotechnology) was also evaluated by immunocytochemistry.

### 2.5. Phalloidin Staining

hOBs were fixed with 4% paraformaldehyde, permeabilized with 0.1% Triton X-100 (Sigma-Aldrich) and then incubated with AlexaFluor 488-Phalloidin (1:500 dilution, cat #A12379; ThermoFisher Scientific). Cells were then counterstained with DAPI (1:1000 dilution, cat # D9542, Sigma-Aldrich). Fluorescence images were captured by using a Zeiss LSM800 Confocal microscope (Carl Zeiss Meditec AG, Oberkochen, Germany).

### 2.6. Bone Matrix Mineralization

Bone mineral matrix deposition was assessed by Alizarin Red staining (ARS; Sigma-Aldrich). After fixing with 10% formaldehyde, hOBs were stained with ARS solution (40 mM, pH 4.2) for 20 min at room temperature (RT) [26]. The quantification was performed at a wavelength of 405 nm using a microplate reader (SpectraMAX 190; Molecular Devices, San Jose, CA, USA).

### 2.7. Bone Markers mRNA Expression

Total RNA isolated with Trizol reagent (ThermoFisher Scientific) was reverse transcribed using the High Capacity cDNA Reverse Transcription Kit (ThermoFisher Scientific). The TaqMan Universal Master Mix II and TaqMan Gene Expression Assay (ThermoFisher Scientific) probes for ALP (Hs01029144_m1), human Runx2 (Hs00231692_m1), human OPN (Hs00959010_m1), human COL1a1 (Hs00164004_m1), human OC (Hs01587813_g1), and β2 microglobulin (B2M, Hs99999907_m1) were used according to the manufacturer’s instructions. Gene expression was assessed with the ABI Prism 7900 Sequence Detection System (ThermoFisher Scientific). The relative gene expression was calculated using the comparative 2^−ΔΔCT^ method.

### 2.8. Bone Markers Protein Expression

hOBs (5 × 10^5^) were analyzed for the expression of the following bone markers: ALP (rabbit monoclonal anti-ALP; 1:1000; cat.# EPR4477, Abcam, Cambridge, MA, USA), Runx2 (rabbit monoclonal anti-Runx2; 1:800; cat.# 12556, Cell Signalling, Danvers, MA, USA), COL1a1 (rabbit monoclonal anti-COL1a1; 1:1000; Abcam, cat.# P02452), OPN (rabbit polyclonal anti-OPN; 1:1000; Abcam, cat.# P10451), and OC (rabbit polyclonal anti-OC; 1:100; Abcam, cat.# O60422) for 30 min at 4 °C. Anti-rabbit Alexa 488 (1:100, Invitrogen, Thermo Fisher Scientific) was used as secondary antibody. Each sample was processed using an FACS Canto II flow cytometer (BD Bioscences) and data were analyzed using FACSDiva v6.1.3, IDEAS software (BD Biosciences) and FlowJo v8.3.3 software (Tree Star Inc., Ashland, OR, USA). Results were expressed as the MFI (mean fluorescence intensity) ratio calculated by dividing the MFI of positive events by the MFI of negative events (MFI of secondary antibody).

### 2.9. hOBs/hOCs 3-D Dynamic (3D-DyC) Co-Culture

The 3-D dynamic culture condition was set up by using the RCCS-4TM bioreactor (Synthecon™, Inc., Houston, TX, USA), with a High Aspect Ratio Vessel (HARV™; Synthecon™, Inc., Houston, TX, USA). hOBs and hMCs isolated from the same patient were combined (2:1 ratio) to create a 3-D culture system [27]. In detail, 1 × 10^6^ hOBs and 0.5 × 10^6^ hMCs were inoculated in 2 mL of HARV filled with 10% FCS DMEM high-glucose; all air bubbles were removed from the culture chamber. Then, HARV was placed into the RCCS-4TM rotary bioreactor, which was allocated in an incubator at 37 °C, in a humidified atmosphere with 5% CO_2_ and the rotation speed was set up at 4 rpm to mimic the ground gravity [20]. After 24 h, bone aggregates were observed, and the vessels were filled with OM alone or in the presence of 10 μM MK-4 (OM+MK-4). OM was replaced twice a week. Following 2 weeks (14 days) of daily treatment with MK-4, cell aggregates were collected, fixed in 4% PFA, embedded in paraffin, and processed for immunohistochemistry.

### 2.10. Immunohistochemistry

Histological sections (5 μm) of cell aggregates were subjected to immunohistochemistry employing the ImmPRESS (Vectorlabs, Burlingame, CA, USA). To this aim, non-consecutive sections were deparaffinized, rehydrated, and enzymatic treated with 1 mg/mL protease K (Sigma-Aldrich). Slides were then immunostained overnight with the primary antibody against osteopontin (OPN; LF-123, a generous gift from Dr. L. Fisher, National Institutes of Health (NIH), Bethesda, USA; rabbit anti-human, 1:100 dilution), in a humid chamber at 4 °C, followed by treatment with Vecstain ABC reagent purchased from Vectorlabs (Burlingame, CA, USA) for 30 min. The reaction was developed using DAB solution (Vectorlabs; Burlingame, CA, USA). The sections were counterstained with hematoxylin, mounted in glycerol, and observed using the Nikon Eclipse 50i optical microscope.

For ARS staining, the sections were deparaffinized and stained with 40 mM ARS solution (pH 4.2) at RT for 20 min, as already reported.

TRAP staining was carried out with the Acid Phosphatase Leukocyte (TRAP) Kit (Sigma-Aldrich no. 386) according to the manufacturer’s protocol. The staining was quantified by a computerized video camera-based image analysis system (NIH, USA ImageJ software) under bright-field microscopy (Nikon Eclipse 50i; Nikon Corporation, Tokyo, Japan). Mean pixel intensity per area was used for the quantification of histological images of osteopontin (OPN). The percentage of positive area was applied for the quantification of ARS and TRAP staining, considering the tears/holes within the matrix of the samples (three replicates per donors were acquired; five sections per sample; *n* = 4 or *n* = 7, respectively).

### 2.11. Statistical Analysis

The median and 1st and 3rd quartile are reported to summarize continuous variables. The Mann-Whitney U test was applied for comparison between subgroups of donors. Correlation network analysis (CNA) is a network analysis technique using a weighted graph to analyze and visualize the strength of pairwise relationships between two variables expressed as a correlation coefficient. The normal distribution of the data was tested with Jarque-Bera. The Spearman rank correlation coefficient was applied to quantify the strength of relation generating the adjacency correlation matrix where nodes are variables. We used a direct acyclic ego-centered graph to visualize the relation between the bone formation marker (ARS) at baseline and after MK-4 administration, expressed as Δ-ARS for eight perfusion parameters: unOC (pg/mL), cOC/unOC, bALP (ng/mL), PINP (pg/mL), PICP (pg/mL), CTX (ng/mL), and NTX (nM). The variability of the Spearman correlation coefficient was reported as 95% confidence intervals derived from the empirical distribution of the estimator calculated through bootstrap resampling techniques. All statistical tests were 2-sided, with a significance level set at *p* < 0.05. Analyses were performed using the R software environment for statistical computing and graphics (v3.4.1; R Studio, Boston, MA, USA) and GraphPad Prism Software Analysis (v6; GraphPad Software company, San Diego, CA, USA).

## 3. Results

### 3.1. Patients and Clinical Parameters

Data available today on the effect of MK-4 on humans come mainly from clinical studies conducted in Asian women of a postmenopausal age. We designed a proof-of-concept study on 10 Caucasian patients with an osteoporosis-related femoral neck fracture. From each patient, we obtained bone specimens and peripheral blood samples for the isolation of osteoblasts and osteoclast precursors, respectively.

All patients’ demographics and clinical parameters are reported in Table 1. In Figure 1, bone formation and bone resorption markers quantified in the serum of enrolled osteoporotic subjects are shown (Figure 1a). These include bone-specific alkaline phosphatase (bALP), procollagen type I N-terminal propeptide (PINP), procollagen type I C-terminal pro-peptide (PICP), and undercarboxylated osteocalcin (unOC), well-known bone formation markers, and NTX and CTX, the common bone resorption markers derived from degradation of type I collagen [22]. Additionally, as an indicator of vitamin K2 status, we also evaluated the cOC/unOC ratio.

As shown in the correlation network analysis (CNA), using the Spearman correlation coefficient between each pair of variables, we quantified the strength of the relation generating the adjacency correlation matrix where nodes are serum markers (Figure 1b). We demonstrated a positive correlation between CTX vs. NTX and PINP vs. bALP, whereas a negative correlation was observed between bALP vs. unOC, CTX vs. PICP, CTX and NTX vs. unOC, and PICP vs. cOC/unOC.

### 3.2. Cell Phenotype Characterization

Before setting up the 3-D autologous bone construct experiments, according to the scheme reported in Figure 2, the cells were separately characterized for their phenotype following a conventional 2-D monolayer culture.

As shown in Figure 3a, cells from bone fragments showed the typical fibroblast-like morphology, were able to deposit bone mineralized matrix (Figure 3b), and expressed the typical bone markers ALP, Runx2, OC, OPN, and COL1a1 (Figure 3c). These properties were also maintained by hOBs after their freezing-thawing (Appendix A).

Regarding osteoclasts precursors, following PBMCs’ adhesion selection on polystyrene plates, purified cells from peripheral blood expressed the specific monocyte membrane marker CD14 (>90%) (Appendix A) and when exposed to the osteoclastogenic inducers RANKL and M-CSF, became TRAP, cathepsin K, and NFAT positive, confirming their ability to differentiate into mature multinucleated hOCs, also after thawing (Appendix A).

### 3.3. hOBs’ Responsiveness to MK-4 in the 2-D Monolayer Cell Culture System

We then focused on osteoblast responsiveness to MK-4. In a first step, 2-D cultured hOBs were exposed to different concentrations of MK-4 (0.1–1–10 μM), without showing significant cytotoxicity (data not shown). In agreement with our previous study [21], we employed MK-4 at 10 μM for all subsequent experiments. In detail, osteo-induced hOBs were exposed to MK-4 daily for 14 days and showed, compared to MK-4-untreated cells, a significant increase in mineral matrix deposition (Figure 4a), as well as in the expression of typical osteogenic markers (ALP, Runx2, OC, OPN, and COL1A1) both at the mRNA (Figure 4b) and protein level (Figure 4c).

### 3.4. MK-4 Effects on hOBs/hOCs 3D-DyC Co-Culture System

hOBs and hMCs from the same patient were then used to create a 3-D culture system. As detailed in the the material and methods section and as previously described [20], cells were subjected to a 3D-DyC dynamic co-culture condition in OM both in the presence and absence of MK-4. After 24 h, the formation of sizeable self-assembled cell aggregates was appreciable (data not shown). Then, 3-D aggregates were collected at 14 days and both types of cell populations were specifically analyzed. Regarding the osteoblastic counterpart, 3D-DyC aggregates exhibited appreciable ARS-positive areas, which, in most cases, significantly increased after MK-4 treatment (Figure 5a). Accordingly, OPN immunohistochemical analysis revealed a significant increase of its expression after MK-4 treatment (Figure 5b). As regards the osteoclastic counterpart, TRAP assay revealed the presence of multinucleated cells mainly in the outer part of the aggregate (Figure 5c), demonstrating the ability of hOBs to support osteoclastogenesis in the absence of osteoclastogenic inducers. Notably, the presence of MK-4 induced a significant decrease of the positive TRAP areas (Figure 5c).

### 3.5. Specific Effect of MK-4 on Each hOBs/hOCs Aggregate and CNA for BTMs

We then focused on the ARS data of each 3-D cell aggregate to study their correlation with the serum levels of the patients’ BTMs from which the cells derived. Since mineralized matrix deposition is a terminal differentiation event, ARS data were chosen for correlation analysis. Furthermore, it is particularly reliable for the evaluation of the effect of specific anti-osteoporosis molecules, thus helping us to move closer to the development of patient-oriented medicine.

As reported in Figure 6a, the effect of MK-4 on each hOBs/hOCs 3-D aggregate was variable. In four out of seven 3-D aggregates analyzed (samples 2, 4, 6, and 7), MK-4 increased the ARS level. In the other three samples (1, 3, and 5), MK-4 did not improve mineral matrix deposition. Nevertheless, the median variation of ARS following MK-4 treatment, expressed as Δ-ARS, was 0.65 (Interquartile Range, IQR: −0.19; 1.77) (Δ ≥ 0; Figure 6b).

To correlate the ARS data obtained in vitro with the serum BTMs of the enrolled osteoporotic patients, a CNA analysis was performed. Figure 6c shows the correlation between basal ARS (percentage of positive area in the 3-D cell aggregates) to the bone markers. A positive relation was found vs. the bone formation markers, such as cOC/unOC (rho = 0.286, 95% Confidence Interval, CI: −0.595; 0.855), bALP ng/mL (rho = 0.071, 95%CI: −0.720; 0.782), and PICP pg/mL (rho = 0.643, 95%CI: −0.214; 0.941). An unexpected negative relation for PINP pg/mL (rho = −0.214, 95%CI: −0.833; 0.642) was observed. Indeed, we observed an expected negative relation to CTX ng/mL (rho = −0.036, 95%CI: −0.768; 0.737), NTX nM (rho = −0.429, 95%CI: −0.893; 0.479), and unOC (rho = −0.179, 95%CI: −0.933; 0.269).

A similar pattern for the same parameters was observed when we correlated BMTs’ values vs. the variation of ARS staining (Δ) after MK-4 treatment in the 3D-DyC aggregates (Figure 6d): unOC pg/mL (rho = −0.751, 95%CI: −0.620; 0.844), cOC/unOC (rho = 0.607, 95%CI: −0.684; 0.809), bALP ng/mL (rho = 0.643, 95%CI: −0.821; 0.664), PICP pg/mL (rho = 0.571, 95%CI: −0.684; 0.809), PINP pg/mL (rho = 0.071, 95%CI: −0.364; 0.918), CTX ng/mL (rho = −0.071, 95%CI: −0.269; 0.933), and NTX nM (rho = 0.071, 95%CI: −0.479; 0.893).

## 4. Discussion

Bone loss and the consequent bone fragility are typical features of osteoporosis, which is one of the major health problems negatively affecting the quality of life post-menopause and in the elderly [28]. Several studies aiming to develop new pharmacological approaches to treat or delay its progression have been performed [2]. Furthermore, lifestyle and nutrition seem to play a key role [29,30].

In this scenario, vitamin K2 was recently identified as a family of promising fat-soluble natural compounds necessary for bone health [31]. Indeed, its capacity to reduce loss of bone mass and risk of fractures thorough their action as a cofactor of the enzyme γ-glutamyl carboxylase (GGCX) [32] has been widely demonstrated [13,14,15,33]. However, the exact role of vitamin K2 in bone disorders, such as osteoporosis, is poorly understood [34]. In particular, it is still unclear if all patients are responsive to its treatment or not. Therefore, due to the high incidence of osteoporosis, the optimization of preclinical tests able to screen potential compounds able to prevent or slow down its progression is needed [34].

It is well established that the employment of bone cell 2-D cultures, as well as in vivo studies, represent the standard approaches to test candidate compounds. However, the lack of adequate bone disease animal models and the need to overcome critical issues, such as ethical concerns, high costs, and low transferability of data to humans [35], is driving the development in vitro of smart 3-D cell culture systems one step closer to natural conditions. Indeed, 3-D cell culture approaches based on the use of healthy or pathological human primary bone cells are very promising because they are able to create in vitro experimental models tailored to the patient [36,37]. In addition, 3-D models could be very useful for investigating drug responsiveness and to optimize therapies according to the principle of patient-oriented medicine.

Based on previous studies performed using 3-D scaffold-free hOBs/hOCs co-cultures [20,38,39], in this study, we demonstrated the feasibility to generate an autologous hOBs/hOCs 3D-DyC system with bone cells from osteoporotic patients undergoing surgery for femoral neck fracture. Of note, despite the low number of available cells, due to the age and clinical conditions of the patients [40], seven sizeable 3-D bone-like aggregates were obtained. Although the low number of patients recruited might be an apparent limit of this study, it is noteworthy to consider this a pilot study whose major strength is the capability of generating an in vitro 3-D model that closely resembles human pathophysiology, which might be used in the future for precision medicine studies. To the best of our knowledge, this is the first demonstration that by using bone cells derived from osteoporotic patients, it is possible to create in vitro 3-D bone constructs to investigate the effects of osteo-bioactive compounds, such as vitamin K2.

Although it the anabolic effect of vitamin K2 on an osteoblast cell line cultured in a 2-D system has been well established [41,42,43,44], recently, Schröder and colleagues [45] investigated the effect of vitamin K2 using human 3-D bone spheroids (osteospheres), which, unlike our approach, were derived from healthy primary hOBs. The hypothesis that the responsiveness of the hOBs/hOCs 3D co-culture system we developed can predict the patient’s responsiveness to a specific treatment is an important challenge that requires further investigation. However, our preliminary data presented here clearly support this assumption and lay the groundwork for additional research.

In detail, by growing hOBs in the conventional 2-D monolayer cell culture system, vitamin K2, used as MK-4, the isoform that is the most abundant in the human body [46], exhibited a general anabolic effect in all samples analyzed (Figure 4). However, when the same hOBs were co-cultured with the corresponding patient’s osteoclast precursors (hMCs), the generated 3-D osteo-aggregates did not maintain the same responsiveness as the 2-D cultured cells. In particular, four samples were responsive to MK-4, while in the other three cases, the treatment was not effective (Figure 6). This could be explained by the fact that patients who probably may be refractory to vitamin K2 treatment in vivo also provide bone cells that reveal their non-responsiveness to this natural compound once cultured in vitro. Therefore, this approach suggests that the use of autologous 3-D co-culture systems may represent an alternative strategy for a specific patient-response screening of anti-osteoporotic compounds in vitro.

In the present study, we also evaluated for each enrolled patient some biochemical parameters of bone metabolism. As expected, the clinical parameters (Table 1) showed that all patients presented T-score values ≤ −2.5 [47], low 25-OH vitamin D levels [48], and an elevated inflammatory state due to the bone injury [49]. In addition, serum levels of BTMs were measured: bALP, PINP, PICP, and OC are deemed important bone formation markers [50]. In particular, OC exists in two circulating forms [8]: undercarboxylated (unOC), used as a bone formation marker because it does not bind hydroxyapatite, and carboxylated (cOC), which, instead, binds bone mineralized matrix [22]. Since total OC is considered a non-specific indicator of bone activity and, therefore, must be interpreted with attention, we measured both circulating forms and their ratio (cOC/unOC) as a predictor of bone formation. Furthermore, regarding bone resorption, the main recommended markers are NTX and CTX; both are fragments released from collagen enzymatic degradation [51]. As expected, as shown by CNA analysis (Figure 1), a positive correlation among CTX vs. NTX and PINP vs. bALP was found. A negative correlation was observed between CTX vs. PICP and CTX and NTX vs. unOC. Whereas, bALP vs. unOC and PICP vs. cOC/unOC were unexpectedly negatively correlated. These data are probably due to several other factors that can affect BTM measurements, such as age, sex, lifestyle, and the occurrence of a fracture [51].

It is known that BMTs are considered valuable bone quality indicators and potential markers to follow response to anti-osteoporotic therapies [52,53]. Thus, for the first time, a correlation between the BMT values of osteoporotic patients and the data obtained in vitro from our 3-D cell-based bone constructs was performed (Figure 6). As expected, the basal levels of ARS in vitro (index of bone mineral matrix deposition) were positively correlated with the main bone formation markers bALP, cOC/unOC, and PICP, while they were negatively correlated with NTX and CTX. However, a negative relation versus PINP and unOC was observed, possibly due to the biological variability of the bone cells derived from different osteoporotic patients. Moreover, by correlating the median variation of bone matrix deposition following MK-4 treatment, expressed as Δ-ARS, we found a positive correlation vs. the bone formation markers (bALP, cOC/unOC, PINP, and PICP), thus suggesting that our autologous 3-D bone construct represents a useful model to test in vitro a patient’s possible response to MK-4.

Overall, our data give proof that co-cultured osteoblasts and osteoclast precursors isolated from the same osteoporotic patient may be suitable to generate an in vitro 3-D experimental model that potentially reflects the individual’s bone metabolism. Since in the 3-D systems, cellular behavior reflects in vivo tissue functionality more accurately than in monolayer cultures, the experimental model proposed here has proven to be an effective platform to screen cells’ ability to respond or not to specific compounds of interest, such as vitamin K2.

In conclusion, our approach represents an innovative step toward patient-oriented medicine since it may be useful to predict how nutraceutical or drug molecules could positively impact bone homeostasis and health.

## Figures and Tables

**Figure 1 nutrients-13-02823-f001:**
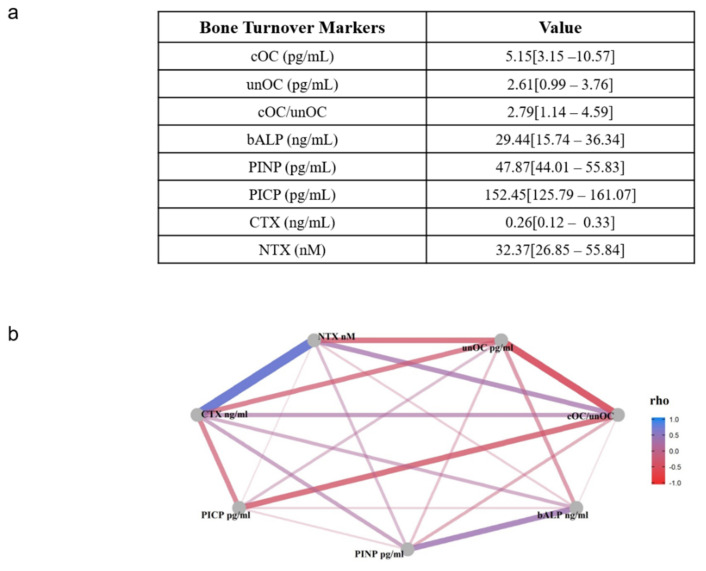
BTM serum levels and correlation network analysis (CNA). Panel (**a**) shows descriptive statistics for continuous patients’ characteristics expressed as the median and 1st and 3rd quartile. Panel (**b**) graphs the correlation network using the Spearman correlation coefficient (rho) between all variables included in the study. Bone-specific alkaline phosphatase (bALP); procollagen type I N-terminal propeptide (PINP); procollagen type I C-terminal pro-peptide (PICP); undercarboxylated osteocalcin (unOC); carboxylated osteocalcin (cOC); cross linked N-telopeptide of type I collagen (NTX); cross linked C-telopeptide of type I collagen (CTX).

**Figure 2 nutrients-13-02823-f002:**
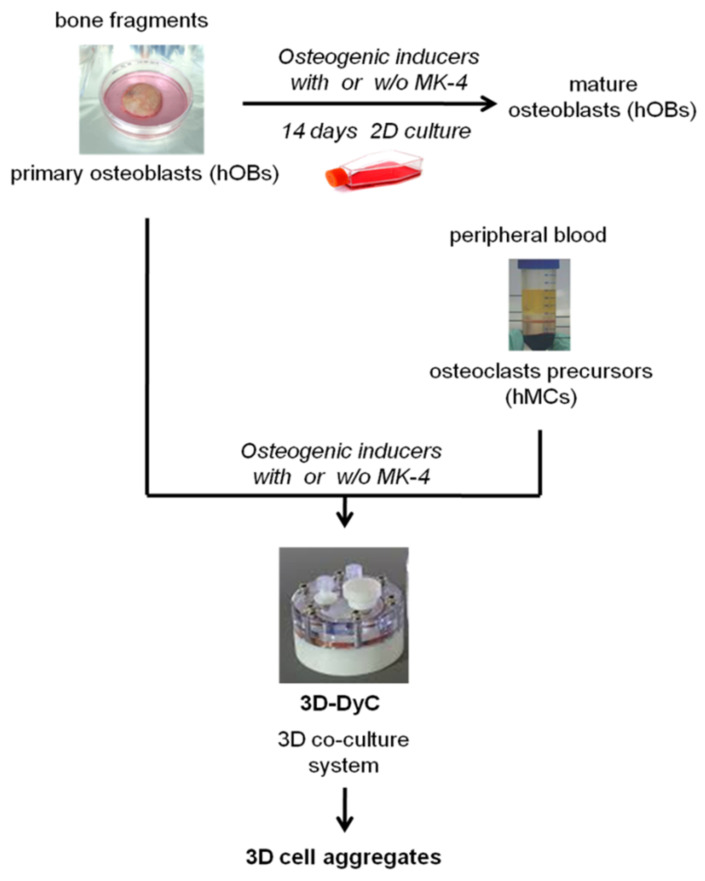
Experimental plan. Bone fragments and peripheral blood samples derived from the same osteoporotic patient were used as sources of osteoblasts (hOBs) and osteoclast precursors (human monocytes, hMCs), respectively. hOBs were treated or not with MK-4 (10 μM) for 14 days, both in a 2-D conventional culture system (upper part of the scheme) and 3-D co-culture with hMCs producing 3-D bone cells aggregates (lower part of the scheme).

**Figure 3 nutrients-13-02823-f003:**
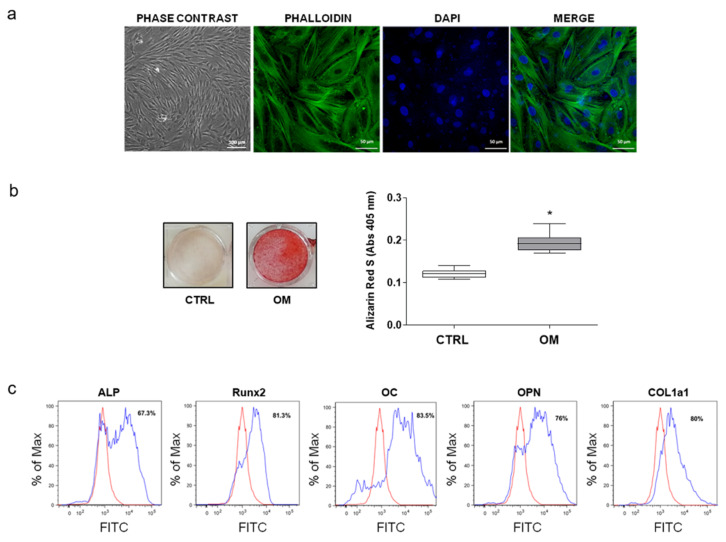
Characterization of hOBs. (**a**) Representative phase contrast and immunofluorescence images of isolated hOBs. (**b**) Mineral matrix deposition was evaluated by ARS. Results are expressed as the median and 1st and 3rd quartile of at least three independent experiments (*n* = 5) (* *p* < 0.05 vs. CTRL). (**c**) Expression of typical osteoblastic markers (ALP, Runx2, OC, OPN, and COL1a1) analyzed by flow cytometry (representative histograms).

**Figure 4 nutrients-13-02823-f004:**
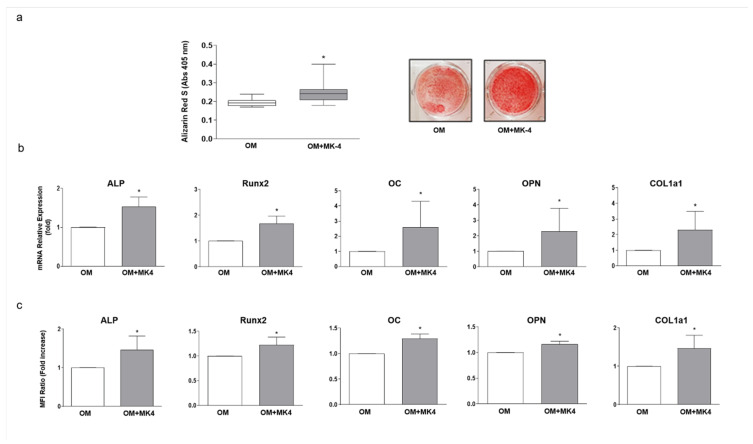
Effect of MK-4 treatment on 2-D cultured hOBs. hOBs were cultured in osteogenic medium (OM) combined or not with MK-4 at 10 μM (OM + MK-4) for 14 days. Mineral matrix deposition was evaluated by ARS and results are expressed as (**a**) the median and 1st and 3rd quartile of at least three independent experiments (*n* = 10). (**b**,**c**) Expression of typical osteoblast markers (ALP, Runx2, OC, OPN, and COL1a1) analyzed both at the mRNA and protein level, by RT-qPCR and flow cytometry, respectively. Data are presented as mean ± standard deviation (SD) of at least three independent experiments (*n* = 5). (* *p* < 0.05 vs. OM).

**Figure 5 nutrients-13-02823-f005:**
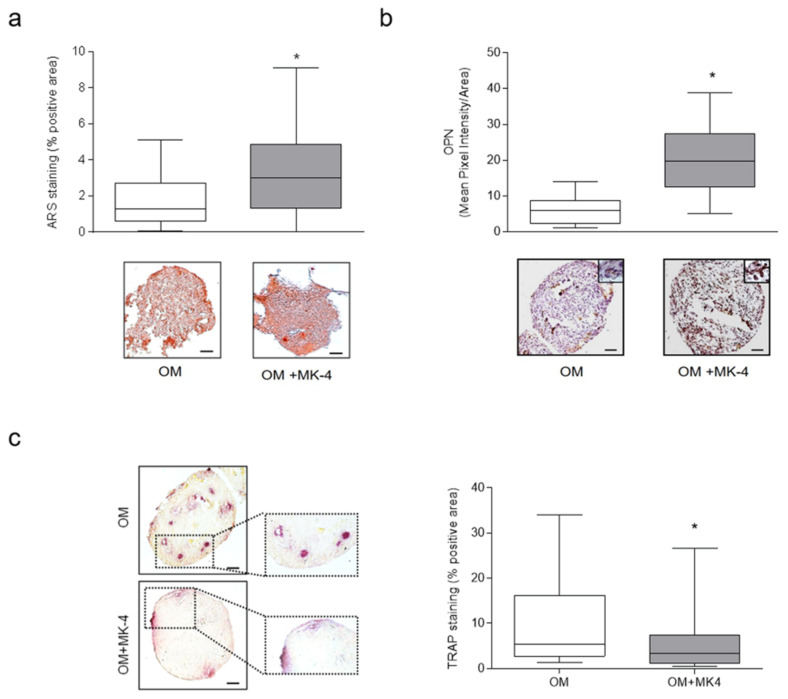
Effect of MK-4 on hOBs/hMCs 3-D co-culture system. hOBs were combined with hMCs and cultured for 14 days in a HARV vessel (dynamic RCCS bioreactor culture system; 3D-DyC condition) in osteogenic medium (OM) alone or combined with MK-4 10 μM (OM + MK-4). (**a**) Representative images of ARS, (**b**) OPN immunohistochemistry expression, and (**c**) TRAP activity are reported. Bars: 50 μm. High magnification images are shown in the insets. The quantification analysis was performed using ImageJ software. Data are presented as the median and 1st and 3rd quartile. (* *p* < 0.05 vs. OM).

**Figure 6 nutrients-13-02823-f006:**
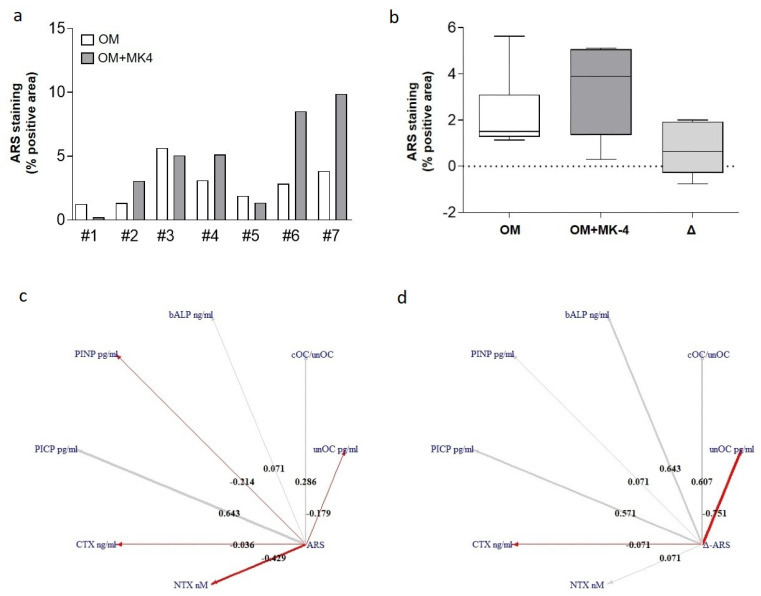
Effect of MK-4 on each hOBs/hMCs aggregates and CNA for BTMs. Panel (**a**) shows the ARS (% positive area) for each sample. Baseline (OM), after MK-4 treatment, and relative differences for ARS (Δ-ARS) are shown as a box-plot (**b**). Ego-centered network for ARS (**c**) and Δ-ARS (**d**), respectively. The weight of the arrow indicates the correlation coefficient of Spearman (rho), the red color indicates a negative correlation, the grey color a positive one, and the thickness of the arrow indicates the strength of the relationship.

**Table 1 nutrients-13-02823-t001:** Demographics and clinical parameters of the patients.

Variable	Value	Normal Range
Gender	8 (80.0%) F; 2 (20.0%) M	
Age (years)		
Protein C reactive (mg/dL)	3.3 (1.5; 5.7)	0–0.5
Serum Ca (mmol/L)	2.1 (2.0; 2.2)	2.1–2.6
Serum P (mmol/L)	1.7 (1.4; 3.4)	0.8–1.5
Urine Ca (mmol/L)	5.7 (2.9; 10.0)	2.5–6.2
Urine P (mmol/L)	21.3 (15.3; 25.5)	2.5–4.5
PTH (pg/mL)	25.7 (19.9; 29.3)	8.7–79.6
25-OH Vitamin D (ng/mL)	22.5 (14.4; 63.5)	31.0–100.0
T-score Femur	−2.7 (−2.8; −2.6)	> −1
T-score Lumbar (L1)	−2.5 (−2.8; 0.4)	> −1
T-score Lumbar (L2)	−2.6 (−2.7; −1.8)	> −1
T-score Lumbar (L3)	−2.9 (−3.0; −1.5)	> −1
T-score Lumbar (L4)	−2.5 (−3.3; −1.1)	> −1
T-score Vertebral	−2.2 (−2.6; −2.0)	> −1

Data are expressed as median (IQR). PTH: parathyroid Hormone, Ca: Calcium, P: Phosphorus. Gender distribution is expressed as absolute frequency (column percentage).

## Data Availability

The data analyzed in this study are not publicly available because individual privacy may be compromised. Interested groups could contact Domitilla Mandatori (domitilla.mandatori@unich.it) to request permission to access these datasets.

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
