# Peer review of "Three-Dimensional Co-Culture System of Human Osteoblasts and Osteoclast Precursors from Osteoporotic Patients as an Innovative Model to Study the Role of Nutrients: Focus on Vitamin K2"

_nutrients, 2021, doi:10.3390/nu13082823_

Round 1
Reviewer 1 Report
This is an interesting paper where the authors show an innovative 3D co-culture system of bone tissue. The authors use this in vitro system to study the role of Vitamin K2 on osteoporotic bone metabolism. Some issues I'd like to see addressed:
As we know there is a big difference in the incidence of osteoporosis in men and women and it is dependent on endocrine factors. The authors used samples from eight women and two men. Wouldn't it be more appropriate to evaluate data separately according to gender? Would the exclusion of the male samples (because are only 2) affect the study results?
I have some doubts regarding Figure 5. The distribution of graphs a and c do not seem compatible with the statistical difference. Is this correct? What statistical methods were used? Also regarding figure 5 B the image of the sample submitted to vit K is not very clear. Do you have a better image to replace this one or a couple more images for review purposes only?
In Figure 6 the graph d is equal to graph c, showing the same correlation values. What is the significance of graph d (variation)?
Line 78 - I believe this is the first time BTM is used but it is not defined.
Line 267 - The first sentence should have a reference.
Line 376 - What do you mean by using the term "assumed". Please clarify.
Author Response
We thank reviewer #1 for the appreciation of our manuscript as well as for his/her useful comments in response to which changes have been made. Particularly, the sections modified or added in the text are highlighted in red as track changes mode.
- As we know there is a big difference in the incidence of osteoporosis in men and women and it is dependent on endocrine factors. The authors used samples from eight women and two men. Wouldn't it be more appropriate to evaluate data separately according to gender? Would the exclusion of the male samples (because are only 2) affect the study results?
We thank the reviewer for his/her correct observation since osteoporosis is a bone disorder commonly associated to women (Almeida M, et al., Estrogens and Androgens in Skeletal Physiology and Pathophysiology. Physiol Rev. 2017; J.A. Kanis et al., European guidance for the diagnosis and management of osteoporosis in postmenopausal women, Osteoporosis International 2019).
Nevertheless, our goal was to develop in vitro a valid 3D bone constructs based on co-culturing bone cells derived from the same osteoporotic patient, which might be potentially useful in the future for precision medicine studies. To this aim, ten osteoporotic patients were randomly enrolled regardless of whether they were male or female. Since osteoporosis is more frequent in women, as happens in reality, we were able to isolate bone cells from eight women and two men. We would like to point out that this is an experimental study and not a clinical trial, in which were investigated the biomolecular and not the epidemiological effects of vitamin K2. Therefore, although the samples size may be appear low, we believed that the number of 3D bone constructs obtained and analyzed could be appropriate for the proposed proof-of-concept study. Furthermore, several difficulties were found in enrolling osteoporotic patients who met the study inclusion criteria to get the appropriate number of autologous bone derived-cells. In this regard, setting up of 3D co-culture experiments required many available cells that, not always, we are able to obtain due to the age and clinical conditions of the patients.
Finally, focusing only on men would lead to very small sample size. In general, the reviewer are right because sex is a very important confounder to control for. However, in our context, we observe that the values for males are not outliers and are very similar to those on women, and thus, including them in the analysis, in this case, they do not alter our results.
- I have some doubts regarding Figure 5. The distribution of graphs a and c do not seem compatible with the statistical difference. Is this correct? What statistical methods were used? Also regarding figure 5 B the image of the sample submitted to vit K is not very clear. Do you have a better image to replace this one or a couple more images for review purposes only?
In order to satisfy the request of the reviewer, we have checked all the quantifications (in terms of positive area or mean pixel intensity/area) and we have added further data that have been included in figure 5 (see page 11). We would like to point out that data are represented as median, 1st and 3rd quartil, in which: a) the ends of the box are the upper and lower quartiles; b) the two lines outside the box extend to the highest and lowest observations; c) the horizontal line inside the box means the median; d) statistical analysis was performed by the Mann-Whitney U test.
Furthermore, in the new version of the figure 5, we have added high magnification images to make more clear the effect of vitamin K2 on osteopontin (OPN) expression levels (insets in figure 5b) (see page 11). For clarity, we also attached a supplemental figure (for the reviewer only) with the same images just after DAB staining, without haematoxylin counterstaining.
- In Figure 6 the graph d is equal to graph c, showing the same correlation values. What is the significance of graph d (variation)?
We thank the reviewer for his/her proper observation. This is a mistake and we apologize for this. In the revised version, we have modified the figure 6 replacing the graph with the correct values for Δ-ARS correlation (figure 6d) (see page 12).
- Line 78 - I believe this is the first time BTM is used but it is not defined
We have added the full name “Bone Turnover Markers” in the revised version (see page 2).
- The first sentence should have a reference.
As suggested by the reviewer, we have added the following reference for the first sentence in the text: Panahi N, Arjmand B, Ostovar A, Kouhestani E, Heshmat R, Soltani A, Larijani B. Metabolomic biomarkers of low BMD: a systematic review. Osteoporos Int. 2021 Jul 26. doi: 10.1007/s00198-021-06037-8. Epub ahead of print. PMID: 34309694. (See page 1).
- Line 376 - What do you mean by using the term "assumed". Please clarify.
With the term “assumed”, we mean an “expected” negative correlation. We have corrected this in the revised version (see page 12).

Reviewer 2 Report
The authors demonstrates an interesting study of co-culture of human osteoblasts and osteoclasts precursors from bone and peripheral blood with 3D system. They reveal that, with this novel model, they can mimic bone formation more precisely and to check the effects of nutrients on bone, such as vitamin K2.
- The number is small and there is differenct result in Figure 6. Though with explanation in discussion section, the authors should have explanation on it.
- There are typos to be corrected, like 2x106 bone cells in line 150, 5x105 hOBs in line 200, etc.
Author Response
We thank reviewer #2 for the appreciation of our manuscript as well as for his/her useful comments in response to which changes have been made. Particularly, the sections modified or added in the text are highlighted in red as track changes mode.
- The number is small and there is different result in Figure 6. Though with explanation in discussion section, the authors should have explanation on it.
We are aware that the sample size may be appear low for this study. This is due to the difficulty to enroll osteoporotic patients who meet the study inclusion criteria to get the appropriate number of autologous bone derived-cells. Indeed, setting up 3D co-culture experiments to generate the 3D osteo-aggregates required many available cells that, not always, we are able to obtain due to the age and clinical conditions of the patients. For this reason, we believed that the number of 3D bone constructs obtained and analyzed could be appropriate for the proposed proof-of-concept study. In addition, we would like to point out that this is an experimental study and not a clinical trial, in which were investigated the biomolecular and not the epidemiological effects of vitamin K2
For what concerns the difference between data, in figure 6a we analyzed the responsiveness to vitamin K2 of each single 3D-osteo-aggregates constructs obtained from the seven osteoporotic patients bone derived-cells. Analysing each of them individually, we found that four were responsive to this natural compound, while in the other three cases it did not induce any effects. This probably might be explained by the fact that these three last ones were composed of patient-derived bone cells, which would probably not be responsive to the vitamin even in vivo. As suggested by the reviewer, we have modified the discussion according to this (see page 13).
- There are typos to be corrected, like 2x106 bone cells in line 150, 5x105 hOBs in line 200, etc.
We have corrected the typos in the revised version.
Reviewer 3 Report
This is a simple study evaluating the effect of vitamin K2 supplementation osteoblastic activity under 2D and 3D culture conditions. In both cases, there is apparent activity, and it is argued that the 3D condition is potentially more relevant because the response is more variable, thereby mimicking the natural condition. However, the sample size is rather limited, the 2D and 3D cultures were not tested in parallel and there is a lack of healthy control cells, or even an osteoblast cell line, for comparison. This is a solid preliminary, or proof of concept study, but is limited in scope for the reasons cited. There is nothing fundamentally wrong, however.
In the methods, it is stated that cells were treated for up to 14 days at times. So, does this mean cells were sometimes treated for less than 14 days? If so, why? This needs to be more clear.
A more thorough discussion of Vitamin K2 would be useful for those that are unfamiliar with this substance. Also, justification for the specific form used would be helpful.
The English grammar needs some work.
Author Response
We thank reviewer #3 for her/his accurate reading of our manuscript and for his/her useful comments in response to which changes have been made. Particularly, the sections modified or added in the text are highlighted in red as track changes.
- This is a simple study evaluating the effect of vitamin K2 supplementation osteoblastic activity under 2D and 3D culture conditions. In both cases, there is apparent activity, and it is argued that the 3D condition is potentially more relevant because the response is more variable, thereby mimicking the natural condition. However, the sample size is rather limited, the 2D and 3D cultures were not tested in parallel and there is a lack of healthy control cells, or even an osteoblast cell line, for comparison. This is a solid preliminary, or proof of concept study, but is limited in scope for the reasons cited. There is nothing fundamentally wrong, however.
We are in agreement with the observations of the reviewer. As cited by his/her, this is a proof-of-concept study, in which the low number of samples analyzed may represent an apparent limit. However, we would like to point out that this is an experimental study and not a clinical trial, in which were investigated the biomolecular and not epidemiological effects of the molecule (vitamin K2). Therefore, although the samples size may appear low, we believed that the number of 3D bone constructs obtained and analyzed could be appropriate for the proposed proof-of-concept study. Furthermore, several difficulties were found in enrolling osteoporotic patients who met the study inclusion criteria to get the appropriate number of autologous bone derived-cells. In this regard, setting up of 3D co-culture experiments required many available cells that, not always, we are able to isolate due to the age and clinical conditions of the patients. Moreover, it is important to specify that each pair of patient-derived bone cells (osteoblasts and osteoclasts’ precursors from the same osteoporotic patients) was analyzed both in 2D and 3D condition.
Finally, we are aware regarding the lack of healthy cells or osteoblast cell line for a comparison. However, in this study our main interest was to develop in vitro a valid 3D cell culture system one step closer to bone microenvironment based on co-culturing osteoblasts and osteoclasts precursors obtained from of same osteoporotic patient, and then to investigate the vitamin K2 effects on this. On the other hand, it was already well established the effect of vitamin K2 on healthy osteoblast (Li W, Zhang S, Liu J, Liu Y, Liang Q. Vitamin K2 stimulates MC3T3 E1 osteoblast differentiation and mineralization through autophagy induction. Mol Med Rep. 2019 May; Zhang YL, Yin JH, Ding H, Zhang W, Zhang CQ, Gao YS. Protective effect of VK2 on glucocorticoid-treated MC3T3-E1 cells. Int J Mol Med. 2017 Jan; Yamaguchi M, Weitzmann MN. Vitamin K2 stimulates osteoblastogenesis and suppresses osteoclastogenesis by suppressing NF-κB activation. Int J Mol Med. 2011 Jan; Katsuyama H, Saijoh K, Otsuki T, Tomita M, Fukunaga M, Sunami S. Menaquinone-7 regulates gene expression in osteoblastic MC3T3E1 cells. Int J Mol Med. 2007 Feb). However, following the reviewer’s suggestion, we added this concept in the discussion (see page 13).
- In the methods, it is stated that cells were treated for up to 14 days at times. So, does this mean cells were sometimes treated for less than 14 days? If so, why? This needs to be more clear.
We apologize for the misunderstanding. Actually, we have treated the cells for 14 days, which represents the shortest time required to induce mineral matrix deposition in human osteoblasts. This aspect has been best explained in the new manuscript version.
- A more thorough discussion of Vitamin K2 would be useful for those that are unfamiliar with this substance. Also, justification for the specific form used would be helpful.
We thank the Reviewer for this appropriate observation. In the revised manuscript, we have added information regarding Vitamin K2 (see pages 2, 12 and 13).
- The English grammar needs some work.
We apologize for the possible grammar mistakes and misspelling and as properly suggested, we performed the language editing thanks our English mother language collaborator.
Round 2
Reviewer 1 Report
Thank you for addressing my questions. Congratulations on your work.
Reviewer 3 Report
I reviewed the initial submission. The authors have adequately addressed my concerns.